# Ostracism and sharing in an intergroup context

Hila Reem[1], Maor Zeev-Wolf [1,2]*

**1** School of Education, Ben Gurion University of the Negev, Beer-Sheva, Israel, **2** School of Brain Sciences and Cognition, Ben Gurion University of the Negev, Beer-Sheva, Israel

* zeevwolf@bgu.ac.il

## Abstract

Previous research suggests that social exclusion is linked to a decrease in individuals' pro-social behavior. However, this effect has not been examined in an intergroup context. We manipulated social acceptance (using the Cyberball game) to examine participants' sharing with ingroup or outgroup members in a minimal group paradigm. Results revealed that when the prospective recipient was a group member who rejected them, socially excluded participants shared less than their socially accepted counterparts. However, when faced with members of an outgroup, socially excluded participants showed similar levels of prosocial behavior as their socially accepted counterparts. Further results suggest that the tendency of socially excluded participants to act in a less prosocial manner toward members of a group that had rejected them was generalized to the group as a whole (including group members with whom there had been no previous interaction). We discuss the theoretical and practical implications of these findings.

## Introduction

The question of what motivates people to act in a prosocial manner—i.e., expending resources for the benefit of others who are not directly related to them—has been of great interest to researchers in recent decades due to its importance for society. Some researchers have focused on personal reward as a motivator—suggesting that helping others produces feelings of warm glow [1, 2] and increases one's well-being [3]. Others have focused on the societal level, suggesting that concern for others can be advantageous because it fosters reciprocity and sustains social groups [4–6]. Indeed, in environments where people expect reciprocity—that is, where they feel a sense of belonging and set out to provide mutual help, support, and care for one another—prosocial behavior occurs more often [7]. Conversely, when people experience social exclusion, their expectations of reciprocity are undermined, and their prosocial behavior diminishes [7–9].

Several studies conducted in school environments have found that children who their peers reject behave in a less prosocial manner than their more socially accepted counterparts [10–12]. This tendency has also been found among adults [7]. In a series of studies, significant reductions have been demonstrated in prosocial behaviors such as donating, volunteering, helping, and collaborating in a *public good* game as a result of social exclusion [7]. However,

**Data Availability Statement:** The data supporting the findings of this project openly available in the Open Science Framework (OSF) repository at https://doi.org/10.17605/OSF.IO/J8Q9A.

**Funding:** The author(s) received no specific funding for this work.

**Competing interests:** The authors have declared that no competing interests exist.

the causes of this reduction in prosocial behaviors, and the context in which it occurs, have yet to be examined.

In the present research, we examined the hypothesis that socially excluded individuals' tendency to act in a less prosocial manner than socially accepted counterparts may be limited to the group or surroundings in which they feel shunned. Furthermore, we surmised that a person's levels of prosociality toward outgroup members who are not involved in that person's social exclusion would not differ significantly between socially excluded and socially accepted individuals. Based on the literature reviewed above, accepted people are expected to be more generous toward ingroup members, from whom they expect more future reciprocity [4, 6]. Excluded people, however, have no such expectations, which may result in lower levels of prosocial behavior on their part toward ingroup members. Indeed, they are likely to harbor negative feelings toward their group members [13], which may further decrease their motivation to act in a prosocial manner. However, when the recipient of the prosocial behavior is an outgroup member, accepted and excluded people are likely to experience similar expectations and emotions—resulting in similar levels of prosociality.

Favoring one's own group (*ingroup favoritism* or *ingroup bias*) is a well-established phenomenon. People tend to favor members of their ingroup over out-group members—as manifested both in their evaluation of others and in their prosocial behavior, such as in the allocation of resources [14–16]. Moreover, ingroup members elicit greater feelings of closeness and responsibility than outgroup members—as well as heightened emotional responses to distress [17–19] and greater obligations to help [20]. However, an intergroup context does not automatically lead to a preference toward the ingroup (i.e., ingroup favoritism). For example, it is less likely to occur when the ingroup is of low status [21, 22] or when the social norm strongly denounces discrimination [23]. In such circumstances, people are less likely to prefer members of their ingroup over outgroup members.

Our hypothesis that socially excluded people's decrease in prosocial behavior would be confined to fellow ingroup members who have rejected them gives rise to the idea that ingroup favoritism may be attenuated by people's sense of belonging to their group or community. This premise is in line with theories that point to people's level of identification with, and sense of belonging to, their ingroup as a key factor in determining their attitudes and behavior toward the group [24, 25]. According to these theories, a high degree of identification with one's ingroup increases positive evaluations and preference of the ingroup (i.e., ingroup favoritism); however, a low degree of identification decreases positive evaluations and preference of the ingroup [26, 27]. Recent findings from a study conducted on fourth-grade school children support the hypothesis that ostracism will reduce prosocial behavior toward the ingroup but not the outgroup due to a decreased sense of belonging [28]. In this study, the authors found a strong positive correlation between popularity (i.e., the number of children who reported playing with each child during school breaks) and prosocial behavior toward the ingroup (i.e., sharing sweets). Namely, less popular children were less likely to share their sweets. The authors concluded that less popular children are those whose sense of belonging to their group is jeopardized. Hence, they are less likely to feel obligated to act prosaically towards their group.

In summary, while previous research has demonstrated the effect of social exclusion in diminishing prosociality, it has not included an intergroup context. As a result, it remains unclear whether the tendency of socially excluded individuals to behave in a less prosocial manner is generalized or limited only to their behavior toward the rejecting group or environment. In the present research, we aimed to address this gap, by examining the effect of social exclusion on prosocial behavior while varying the recipient's affiliation (ingroup vs. outgroup members). Replicating previous research, we expected an overall ingroup bias (greater sharing

with ingroup than outgroup recipients), and that ostracism would decrease prosocial behavior overall [7]. Importantly, we expected the effect of ostracism on prosociality to occur only when the recipient is a fellow member of one's rejecting group. When the recipient is an outgroup member, who is not a part of the rejecting group, we expected no difference between the sharing propensity of excluded and accepted individuals.

We examined these predictions in two experiments involving the Cyberball paradigm [29, 30]—a virtual ball game designed to manipulate ostracism in a small group—and the minimal group paradigm [31, 32], in which participants are assigned to groups based on an arbitrary criterion. In both studies, participants either did or did not experience social exclusion by their ingroup. They were then asked if they were willing to share resources with another participant who was either an ingroup or an outgroup member. In Study 1, the recipient of the prosocial behavior in the ingroup condition was a fellow participant with whom they interacted earlier in the Cyberball game, while in Study 2, it was a member of their broader group with whom they had had no interaction.' In both studies, all measures, manipulations, and exclusions are disclosed, as well as the method of determining the final sample size. No data was collected after analysis.

## Study 1

The first study was designed to test our main hypothesis—namely, that socially excluded people behave in a less prosocial manner than their socially accepted counterparts within the group that has rejected them, but not in relation to members of an outgroup. The study includes a 2 (Ostracism: Exclusion vs. Inclusion) X 2 (Recipient's Group: Ingroup vs. Outgroup) design. Ostracism was manipulated using the Cyberball game paradigm. The recipient's group affiliation was based on a minimal group procedure. Finally, prosocial behavior was measured by participants' sharing decisions in a dictator game.

### Method

The research has been carried out in accordance with the ethical code of the American Psychological Association (APA). The protocol for this study has received the approval of the School of Education ethics committee at Ben Gurion University of the Negev (reference no. 0220) after carefully considering the study's rationale, procedure, and materials.

**Participants.** To determine the number of participants to recruit for the study, we conducted a power analysis by means of the G*Power computer application [33]. This indicated that a sample of approximately 250 people would be sufficient to detect a small-to-medium effect size ($f = 0.25$) with a power of 90%. A sample of 258 MTurk workers took part in the study via their MTurk user accounts on their own computers.

### Procedure

After signing a digital informed consent form (approved by the School of Education ethics committee; reference no. 0220), participants were invited to take part in a two-part online experiment—a mental visualization task followed by a decision-making task—for a nominal fee of 50¢ and an entry into a raffle, in which five participants would win an extra $5.

**Group assignment.** Once their participation was approved, participants were told that in the first part of the experiment, they would take part in a virtual ball game designed to examine mental skills as a member of one of the teams playing online at that time. Participants saw an image of two teams of three players each, labeled "Blue" and "Yellow." They were told that all participants in the study were being assigned to either of the two teams based on their artistic preferences.

Participants were then presented with three pairs of images and asked to indicate which one they preferred in each instance. This procedure was based on the *minimal group paradigm* [31, 32], which states that, under certain conditions, merely being assigned to an experimental group is enough to instill a sense of belonging to that group [34]. After completing the artistic preference test, all subjects were told that, based on their preferences, they were being assigned to the Blue group and would next join one of the Blue teams currently playing online.

**Cyberball game.** Participants were presented with the *Cyberball* game—a virtual game used to manipulate degrees of social inclusion or exclusion [29, 30]. Within this game, participants were randomly assigned to a social exclusion or inclusion condition. During the game, they were asked to toss a ball back and forth with two other online players from the Blue team —unbeknownst to them. However, the other two players were generated and controlled by the computer program. To disguise the manipulation, participants were asked to visualize the situation, themselves, and the other players, as a means of engaging their mental visualization skills. On receiving the ball, participants could choose which group member to pass it to, with no time limits. All games were designed to involve a total number of 30 ball tosses, and the time between each ball toss was held constant in both conditions.

Based on a meta-analysis of 120 Cyberball studies [35] and a pilot study of our own, we programmed the game such that participants who had been assigned to the Exclusion condition received the ball only twice early on in the game. In contrast, participants in the Inclusion condition received the ball an equal number of times as the other two "players" throughout the game.

Once the game was concluded, participants answered three questions aimed at validating the manipulation, based on previous research [35]: (1) *How much did you enjoy playing the ball tossing game*? (2) *How much did you feel a part of the group*?, and (3) *How much did you feel other players in the group appreciated you as a person*? Participants rated each sentence on a 7-point scale, ranging from *1 (Not at all)* to *7 (Very much)*. The purpose of these questions was to confirm that participants assigned to the exclusion condition of the ball tossing game enjoyed it less and felt less a part of their group and less appreciated by the other players in their group compared to participants assigned to the inclusion condition. Namely, we expected participants in the exclusion condition to have a lower mean score on the three manipulation check questions than participants in the inclusion condition.

**The dictator game.** In the second part of the experiment, participants were asked to take part in a decision-making game, which offered the chance to win an additional $5 on top of their initial 50¢ participation fee. We used the Dictator Game—a well-known paradigm for measuring prosocial behavior. The instructions for the game were as follows:

> In this game participants are assigned to play one of two roles: Player A gets the chance to participate in a raffle and win $5, while player B does not. If player A wins the raffle, he/she will be able to share their win with player B.

Participants were informed that they were being assigned their role at the game at random and that the identity of all players would be concealed. They were all then told (individually) that they had been randomly chosen to play the role of Player A (since we were only interested in the Allocator's decision in this study). Next, they were asked whether they would be willing to share any part of their $5 prize (should they win it in the raffle) with Player B (who did not have a chance to participate).

To manipulate the recipient's group affiliation, participants were randomly assigned to one of two conditions. In the *Ingroup* condition, they were told that Player B—with whom they could share their win—had been a fellow member of their Blue team in the ball-tossing game they had played earlier. In the *Outgroup* condition, they were told that Player B was a member

of the Yellow team, which had played with a different group of players in the ball-tossing game. They were then asked to indicate how much of their $5 prize (if they were to win it) they would be willing to share with Player B—on a scale of $0–$5, at $0.50 increments.

On completing the second part of the experiment, participants were asked to rate their perceptions of their fellow Ingroup members and Outgroup members, on a 7-degree scale, in terms of three positive (*caring*, *kind*, and *friendly*) and three negative attributes (*selfish*, *evil*, and *indifferent to one another)*. Finally, before concluding the study, participants answered a short demographic questionnaire about their age, gender, and native language.

## Results and discussion

**Manipulation check.** Cronbach's alpha of the three manipulation check questions was .952—indicating high internal validity. Accordingly, the mean of the three questions was used to examine the effect of the inclusion/exclusion manipulation on the participants' experience during the game. Results of an independent t-test revealed a significant difference between the two conditions: $t_{(1,256)} = 11.39$, $p < .001$, $d = 1.39$ [CI: 2.07; 2.94]. As expected, excluded participants reported enjoying playing the Cyberball game less, felt a weaker sense of belonging to the group, and felt less appreciated by the members of their group (M = 2.87, SD = 2.03) than those included (M = 5.38, SD = 1.46).

**The amount of money shared.** The amount of money shared by the participants as a function of Ostracism (Exclusion or Inclusion) and the Recipient's Group (Ingroup or Outgroup) is presented in Fig 1. These amounts ranged from nil (29.5%) to $5 (3.1%), (M = .95, SD = 1.14). Results of an ANOVA of the amount shared by Ostracism and the Recipient's Group revealed a significant main effect for Ostracism—$F_{(1,254)} = 12.33$, p = .001, $\eta_p^2 = 0.046$ —such that participants in the Inclusion condition shared more of their $5 prize (M = 1.19, SD = 1.36), [CI: 0.99; 1.37] than those in the Exclusion condition (M = .70, SD = .78), [CI: 0.51; 0.89]. The effect of the Recipient's Group approached significance—$F_{(1,254)} = 3.52$, p = .062, $\eta_p^2 = 0.01$—such that, overall, participants tended to share more with Ingroup recipients (M = 1.07, SD = 1.39), [CI: 0.88; 1.26] than with Outgroup ones (M = .81, SD = .77), [CI: 0.62; 1.00]. Most notably, for the purpose of our study, the interaction between the two factors was

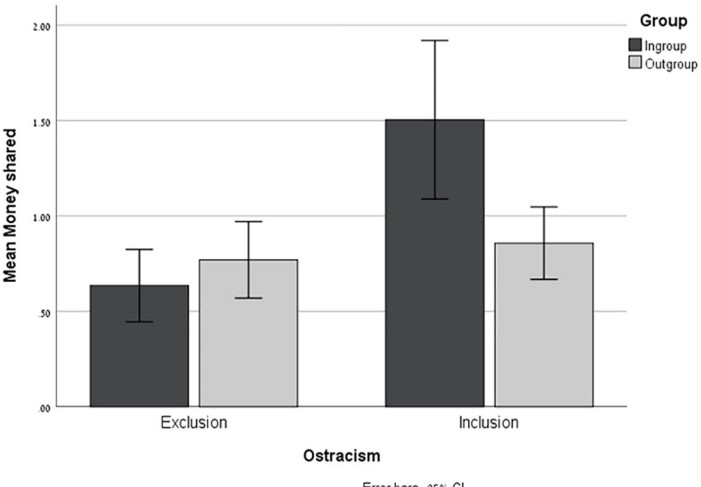

**Fig 1. Study 1: Money shared.** The amount of money shared with Ingroup and Outgroup recipients as a function of the Ostracism manipulation. Error bars represent a 95% confidence interval (CI).

highly significant—$F_{(1,254)}$ = 8.24, $p$ = .004, $\eta_p^2$ = 0.03. Simple effect analysis reveals that in the Ingroup condition, participants who had been included shared significantly more (M = 1.50, SD = 1.70), [CI: 1.24; 1.77] than those who had been excluded (M = .63, SD = .76), [CI: 0.37; 0.90], $F_{(1,254)}$ = 20.84, $p$ < .001, $\eta_p^2$ = 0.08. Conversely, in the Outgroup condition, no significant difference was found between the amount shared by the included (M = .86, SD = .75), [CI: 0.59; 1.13] and excluded participants (M = .77, SD = .80), [CI: 0.50; 1.04], $F_{(1,254)}$ = .20, $p$ = 0.65, $\eta_p^2$ = 001. However, in both conditions, the amount shared was significantly more than nil [$t_{(129)}$ = 9.94, p < .001, and $t_{(127)}$ = 10.15, p < .001 in the Inclusion and Exclusion conditions, respectively]—thus ruling out the possibility of a floor effect.

**Group perceptions.**    Positive ($\alpha$ = .98 and $\alpha$ = .96 for Ingroup and Outgroup respectively) and negative ($\alpha$ = .80 and $\alpha$ = .84 for Ingroup and Outgroup respectively) perceptions of both groups were calculated as the mean of their respective constituent items. Results of independent t-tests revealed that participants in the Inclusion condition rated their group significantly more positively (M = 5.11, SD = 1.52), $t_{(265)}$ = 10.51, $p$ < .001, $d$ = 1.30, [CI: 1.84; 2.96] and less negatively (M = 3.25, SD = 1.96), $t_{(265)}$ = -5.81, $p$ < .001, $d$ = .72, [CI: -1.65; -0.81], than those in the Exclusion condition (M = 2.84, SD = 1.93, and M = 4.48, SD = 1.38; for positive and negative perceptions, respectively). Positive perceptions of the Outgroup were significantly higher in the Inclusion condition (M = 4.52, SD = 1.49) than in the Exclusion one (M = 1.14, SD = 1.22), $t_{(256)}$ = 2.27, $p$ = .024, $d$ = .28, [CI: 0.05; 0.72]. However, negative perceptions of the Outgroup did not significantly differ between the two conditions (M = 3.74, SD = 1.72 and M = 3.80, SD = 1.27 for the Inclusion and Exclusion conditions respectively), $t_{(256)}$ = -.34, $p$ = .74, $d$ = .04, [CI: -0.43; 0.31]. Thus, it appears that while the exclusion manipulation affected the positive perceptions of both groups, negative perceptions were affected only in the case of the Ingroup.

The results of this study replicate those of earlier research—namely that, overall, ostracism diminishes prosocial behavior. However—in line with our hypothesis—this tendency was found to be significant only when the recipient was a fellow member of the rejecting ingroup. When the recipient was an outgroup member and therefore not a part of the group that the participants had felt shunned by, the participants were willing to share their endowment to the same degree as those in the Inclusion condition.

One important question arising from the results of Study 1 is whether the excluded participants' tendency to share less with ingroup recipients (compared with included participants) was limited to the specific team members from their ingroup who had excluded them earlier or whether it would extend to the entire ingroup (i.e., to all members of the Blue group).

To answer that question, we conducted Study 2, in which all participants were assigned to the Ingroup condition (i.e., participants were asked in the end to share with a member of the Blue group), while the prospective recipient was described either as one of the two players that the participant had played with earlier or of the wider Blue group taking part in the study. Thus, the study involved a 2X2 (Ostracism: Inclusion vs. Exclusion X Recipient Type: Team-member vs. Group-member) design. Team member refers to the condition where participants had to share money with one of the two players they directly interacted with from the ingroup. In contrast, Group member refers to the condition in which participants had to share money with another ingroup member they did not directly interact with.

## Study 2

### Method

The research has been carried out in accordance with the ethical code of the American Psychological Association (APA). The protocol for this study has received the approval of the School

of Education ethics committee at Ben Gurion University of the Negev (reference no. 0220) after carefully considering the study's rationale, procedure, and materials.

**Participants.** Based on the effect size found in Study 1, a power analysis by means of the G*Power computer application [33] indicated that a sample of approximately 270 people would be sufficient, with a power of 90%. Accordingly, a sample of 274 MTurk users that did not participate in Study 1 was used (134 women, 139 men, and one unspecified), aged between 19 and 69 (M = 36.22, SD = 11.17). As in Study 1, all participants entered the survey with their MTurk usernames on their own computers.

**Procedure.** The study's procedure was identical to the one described in Study 1: after signing a digital informed consent form (approved by the School of Education ethics committee; reference no. 0220), participants took an "artistic preference test" and were all assigned to the Blue group, ostensibly based on their results. Next—as in Study 1—they played the Cyberball game under one of the two ostracism conditions (inclusion or exclusion) and answered the three manipulation check questions. In the second part of the study, participants were introduced to the Dictator game, in which they had to decide whether and how much money to share with a fellow member of the Blue group, who was either a member of the team the participant had played with earlier (hereafter, a *Team-Member*), or of one of the other Blue teams in the study (hereafter *Group-Member*). Finally, they were asked how they perceived their ingroup and outgroup, using the same six items used in Study 1. However, while participants in the Team-Member condition were asked about the specific members of their team, those in the Group-Member condition were asked about their perceptions of the Blue group as a whole.

## Results and discussion

**Manipulation check.** Cronbach's alpha of the three manipulation check questions was .94. Results of an independent t-test revealed a significant difference between the two conditions, $t_{(1,272)}$ = 16.29, $p < .001$, $d$ = 1.96, [CI: 2.54; 3.24]—such that excluded participants reported enjoying playing Cyberball less felt a weaker sense of belonging to the group, and felt less appreciated by the members of their group (M = 2.18, SD = 1.41) then included participants (M = 5.07, SD = 1.52).

**The amount of money shared.** The mean amount of money shared by the participants as a function of Ostracism (Exclusion vs. Inclusion) and Recipient Type (Team-member vs. Group-member) are presented in Fig 2. The amounts shared ranged between nil (38%) to $5.00 (1.1%), (M = 1.11, SD = 1.15). Results of an ANOVA of the amount shared by Ostracism and Recipient Type revealed a significant main effect for Ostracism—$F_{(1,267)}$ = 32.92, $p < .000$, $\eta_p^2$ = 0.11—such that participants in the Inclusion condition shared more money (M = 1.49, SD = 1.24), [CI: 1.31; 1.68] with Player B than participants in the Exclusion condition (M = .73, SD = .91), [CI: 0.55; 0.92]. No other effect was found to be significant. Specifically, the interaction in question fell far short of significance, $F_{(1,267)}$ = .22, $p$ = .64, $\eta_p^2$ = .001.

**Group perceptions.** As in Study 1, positive perceptions of Ingroup members (α = .98 and α = .94 for Ingroup and Outgroup, respectively) and negative ones (α = .65 and α = .71 for Ingroup and Outgroup, respectively) were calculated as the mean of their three respective constituent items. Results of independent t-tests revealed that participants in the Inclusion condition rated their group members significantly more positively (M = 4.92, SD = 1.43), $t_{(272)}$ = 15.37, $p < .001$, $d$ = 1.85, [CI: 2.29; 2.96], and less negatively (M = 2.53, SD = 1.38), $t_{(272)}$ = -10.60, $p < .001$, $d$ = 1.28, [CI: -1.99; -1.37], than participants in the Exclusion condition (M = 2.30, SD = 1.39 and M = 4.21, SD = 1.23; for positive and negative perceptions respectively). Importantly, these differences remained significant in both the Team-Member

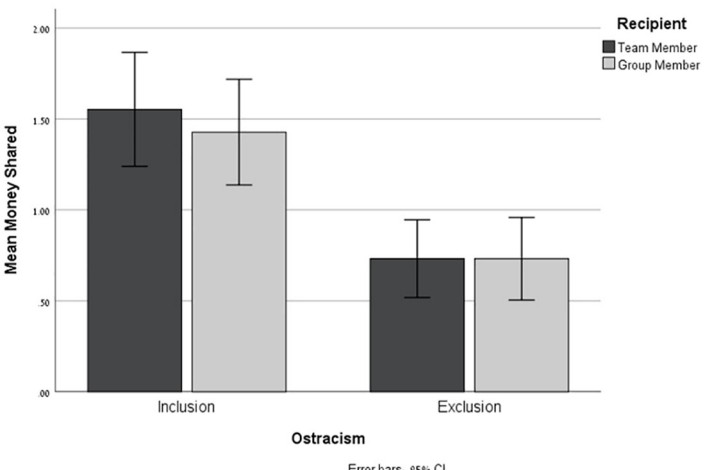

**Fig 2. Study 2: Money shared.** The amount of money shared with Team-Members and Group-Members as a function of the ostracism manipulation. Error bars represent a 95% confidence interval (CI).

(positive perceptions, $t_{(135)}$ = 9.70, $p < .001$, $d$ = 1.66 [CI: 1.85; 2.79] and negative perceptions, $t_{(135)}$ = -6.45, $p < .001$, $d$ = 1.11, [CI: -2.05; -1.09])) and in the Group-Member (positive perceptions, $t_{(135)}$ = 12.09, $p < .001$, $d$ = 2.06, [CI: 2.45; 3.41] and negative perceptions, $t_{(135)}$ = -8.60, $p < .001$, $d$ = 1.47, [CI: -2.18; -1.37])) conditions.

Included and excluded participants' perceptions of the Outgroup did not significantly differ (positive $t_{(272)}$ = .48, $p$ = .63, $d$ = .06, [CI: -0.21; 0.33], negative $t_{(272)}$ = -1.88, $p$ = .06, $d$ = .22, [CI: -0.54; 0.11]). These differences remained insignificant in both Team-Member and Group-Member conditions.

The results of Study 2 replicate the effect of ostracism on prosocial behavior—demonstrating that participants in the Inclusion condition shared significantly more of their endowment with Player B than participants in the Exclusion condition. Notably, this tendency was significant both when the recipient was a fellow ingroup team member with whom the participant had interacted earlier (replicating the results of Study 1), and when the recipient was a member of the wider ingroup with whom the subject had had no previous interaction. This finding suggests that reactions toward specific ingroup team members are extended to the entire ingroup —including those with whom the subjects had had no previous interaction. This pattern was repeated in the participants' perceptions of their team and fellow group members: both positive and negative perceptions of specific team members extended to the participant's perceptions of the group as a whole.

## General discussion

The results of the present study provide new insights into the understanding of the effect of ostracism on prosocial behavior. In line with previous research, our results indicate that ostracism significantly impairs prosocial behavior [7–9, 12]. However, this tendency was found only when the prospective recipient belonged to the rejecting group (Study 1). When the recipient was a member of another group, the socially excluded participants shared to a similar extent as their socially accepted counterparts. However, the results of Study 2 suggest that excluded people's tendency to act in a less prosocial manner toward members of the group who had actively shunned them earlier extends to the group as a whole (including members with whom they had had no previous interaction).

In general, we found that both positive and negative perceptions and emotions toward specific Ingroup team members are extended to the entire ingroup. This finding adds to the literature on negative ingroup stereotypes, which state that people's encounters with a deviant ingroup member may affect how they perceive the ingroup in different ways, depending on the degree to which they identify with the ingroup [15, 24, 32]. People who perceive the deviant person's behavior as being unique to that individual draw a distinction between that person and the group as a whole, thus maintaining a positive perception of the ingroup. However, weak identification with the ingroup (as in the case of social exclusion) may induce people to extrapolate the deviant behavior to the group as a whole and to form negative stereotypes of the group [36].

Previous research on the effect of social exclusion on prosocial behavior [7, 9] did not include such an intergroup setting, and the context in which the behavior was examined was linked to some degree to the exclusion situation in question. For example, in one study, prosocial behavior included assisting the experimenter who was conducting the study in which they had been shunned or donating to a student's fund after being rejected by other students in the study [7]. Our results suggest that excluded people's tendency to act in a less prosocial manner toward individuals who had actively shunned them extends to all members of those individuals' group. Thus, it is possible that participants in previous studies perceived the prospective recipient as being part of the group that had excluded them.

In addition, the results of the current study contribute to the literature on ingroup bias [14–16] by showing that it may also be a function of one's sense of belonging to that group. Thus, while participants in the Inclusion condition shared significantly more with members of their own group (specific ingroup members with whom they interacted earlier, as well as members of the wider ingroup), participants in the Excluded condition showed no such preference. Previous research suggests that people's level of identification with, and sense of belonging to, their ingroup is a key factor in determining their attitudes and behavior toward the group [24, 25]. High levels of identification with the ingroup tend to result in more positive ingroup evaluations and ingroup bias [26, 27]—while being rejected by one's group is thought to reduce self-identification with the group and reduce ingroup favoritism. This finding is in line with the previously noted recent research on situations that may attenuate ingroup bias—such as perceptions of the ingroup's behavior [37], group status [21, 22], and social norms regarding discrimination [23].

Our findings highlight the possibility that the tendency of socially excluded people to act in a less prosocial manner may not always be due to intrinsic factors such as a lack of social skills [38] but rather to exclusion—which decreases identification with the group in question and perceived closeness with its group members [15, 24]. Being excluded can also lower people's expectations of reciprocity from fellow group members [7].

Our study's findings may also suggest practical ways of increasing people's prosociality by encouraging their participation in diverse social groups (e.g., by taking part in several working teams in their work environment or by joining various classes at the community center, etc.). Such diversity in people's social surroundings may enhance their sense of belonging, at least in some of these groups, thereby increasing prosociality within those groups [12, 39, 40]. This may create a "positive loop," in which prosociality promotes future social acceptance, which in turn increases levels of acceptance and prosociality in other groups that the individual belongs to. As our findings demonstrate, the effect of exclusion on prosociality is confined to the excluding group, so belonging to more than one group gives people the opportunity to continue behaving prosocially, even in the face of social exclusion by one particular group.

This idea may be especially relevant in education, as it highlights the importance of encouraging children to take in a wide range of social groups (such as art and music classes and sports

teams, which draw children together from various grades and classes). This would enable them to establish a different social status in each group, which in turn may encourage them to develop diverse behavioral responses in general, and in the prosocial context in particular.

In summary, in this report, we demonstrated a novel social phenomenon regarding the effect of ostracism on prosocial behavior. We provided initial support to the idea that the effect of ostracism on prosociality may be dependent on situational factors, particularly on the group affiliation of the prospective sharing recipient. The research was based on online experiments that utilized the Cyberball game. In the future, we intend to further examine this pattern of results by conducting face-to-face experiments in the lab in order to increase the external validity and generality of the results. Moreover, here we operationalized group affiliation using the minimal group paradigm. Future studies should also seek to replicate our findings in real-world groups to increase the ecological validity of our results. We also intend to use a broader range of ostracism manipulations (e.g., fake sociometric feedback) and examine a variety of prosocial behaviors (such as donating, volunteering, helping, and collaborating). It should be stressed, however, that in today's world, acts of ostracism are increasingly conducted online. Hence, our initial focus on an online research paradigm effectively simulates much of peoples' real-world experience of ostracism as it affects their day-to-day behavior.

## Acknowledgments

We want to thank Prof. Tehila Kogut from Ben Gurion University of the Negev for her priceless contribution to this study.

## Author Contributions

**Conceptualization:** Hila Reem, Maor Zeev-Wolf.

**Data curation:** Hila Reem, Maor Zeev-Wolf.

**Formal analysis:** Maor Zeev-Wolf.

**Methodology:** Maor Zeev-Wolf.

**Supervision:** Maor Zeev-Wolf.

**Writing – original draft:** Hila Reem.

**Writing – review & editing:** Maor Zeev-Wolf.

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
