## [Decision Letter · Decision Letter 0]

14 Apr 2023

PONE-D-23-00987Ostracism and sharing in an intergroup contextPLOS ONE

Dear Dr. Zeev-Wolf,

Thank you for submitting your manuscript to PLOS ONE. After careful consideration, we feel that it has merit but does not fully meet PLOS ONE’s publication criteria as it currently stands. Therefore, we invite you to submit a revised version of the manuscript that addresses the points raised during the review process.

We look forward to receiving your revised manuscript.

Kind regards,

Arslan Ayub

Academic Editor

PLOS ONE

Journal Requirements:

2. Please change "female” or "male" to "woman” or "man" as appropriate, when used as a noun (see for instance https://apastyle.apa.org/style-grammar-guidelines/bias-free-language/gender).

3. Please provide additional details regarding ethical approval in the body of your manuscript. In the Methods section, please ensure that you have specified the name of the IRB/ethics committee that approved your study.

4. Please provide additional details regarding participant consent. In the ethics statement in the Methods and online submission information, please ensure that you have specified (1) whether consent was informed and (2) what type you obtained (for instance, written or verbal, and if verbal, how it was documented and witnessed). If your study included minors, state whether you obtained consent from parents or guardians. If the need for consent was waived by the ethics committee, please include this information.

Reviewers' comments:

Reviewer's Responses to Questions

**Comments to the Author**

1. Is the manuscript technically sound, and do the data support the conclusions?

Reviewer #1: Partly

2. Has the statistical analysis been performed appropriately and rigorously? 

Reviewer #1: Yes

3. Have the authors made all data underlying the findings in their manuscript fully available?

Reviewer #1: No

4. Is the manuscript presented in an intelligible fashion and written in standard English?

Reviewer #1: Yes

5. Review Comments to the Author

Reviewer #1: Thank you for the very interesting paper. I enjoyed reading it particularly, going through the thought process of the researchers. The experiment that were carried out were neat in terms of addressing their objectives. However, I do have a few clarifications that need to be addressed:

1. Line 46, reference missing for the studies referring to adults behaving in a less prosocial manner when rejected by society.

2. Page 4, line 74 to 78 is not clear to me. Please simplify this sentence or rephrase it.

3. “According to these theories, a high degree of identification 81 with one's ingroup increases positive one's ingroup evaluations and ingroup bias”, This sentence is unclear. Perhaps there is a slight mistake on the usage of the word “positive”.

4. “Our hypothesis that socially 75 excluded people's decrease in prosocial behavior would be confined to fellow ingroup 76 members who have rejected them gives rise to the idea that ingroup favoritism may be 77 attenuated by people's sense of belonging to their group or community.” This argument has to be clearly presented and stregthen in the manuscript. As of now, the argument of having an intergroup context is weak.

5. “participants are assigned to groups based on an arbitrary criterion.” It would be nice to use existing ingroup and outgroup criterion, for example ethinicity, age as examples.

6. “(1) How much did you 165 enjoy playing the ball tossing game? (2) How much did you feel a part of the group?, 166 and (3) How much did you feel other players in the group appreciated you as a person?” Please explain how these questions are vaidating the manipulation of the games? If the were to answer 7 – very much, what would it mean and 1 not at all, what would that mean in terms of the manipulation of the game?

7. How can you be sure that the participants in Study 2 are not the same as in Study 1 from MTurk? Could it be a potential bias to the results if same participants signed up for both studies?

8. It would be neat to have two more studies replicating Study 1 and Study 2 in a different context or using a different game. This is to show robustness. In my opinion, two studies are not enough to show the robustness of the results.

9. What is the difference between the usage of words: ingroup vs outgroup and team member vs group members? Can’t they all be ingroup vs outgroup? This brings me to the next question, the comparison between ingroup and outgroup and being a team member in a bigger group is a bit confusing. This should be explain clearer.

10. Please be consistent with the usage of the word Fig vs Figure in Figure caption.

11. The manuscript is written well, the idea is also well thought of. However, the manuscript could improve with the help of a proofreader or streamlining the explanation in a clearer manner. This is important when the studies are complicated.

6. PLOS authors have the option to publish the peer review history of their article (what does this mean?). If published, this will include your full peer review and any attached files.

Reviewer #1: No

---

## [Author Response · Author response to Decision Letter 0]

4 May 2023

Dear Dr. Arslan Ayub,

Thank you and the reviewer for your feedback and careful consideration of our manuscript. We found the reviewer’s comments and suggestions to be very thoughtful and helpful, and have made every effort to implement them in the attached revision. We are confident that the revised paper has been improved by the review process. Below, we have reviewed the changes we have made, vis-à-vis the reviewer and your comments, which are now integrated within the revised manuscript.

Sincerely yours,

Maor Zeev-Wolf

 

Editor’s Comments

Response: Further to the links provided by the editor, the manuscript now meets PLOS ONE’s style requirements, including file naming. 

2. Please change "female” or "male" to "woman” or "man" as appropriate, when used as a noun.

Response: Per the request, we have changed female and male to woman and man (see p.13, line 309, in the revised manuscript).

3. Please provide additional details regarding ethical approval in the body of your manuscript. In the Methods section, please ensure that you have specified the name of the IRB/ethics committee that approved your study.

Response: Additional details were added to the manuscript regarding ethical approval, including the name of the ethics committee (in the Methods section) and the reference number of the approval for the study. Namely, we now explain that the research has been carried out in accordance with the ethical code of the American Psychological Association (APA) and that the protocol for this study has received the approval of the School of Education’s ethics committee at Ben Gurion University of the Negev (reference no. 0220) after it carefully considered the study’s rationale, procedure, and materials (see p.6, lines 134-137, and p.13, lines 301-304, in the revised manuscript). 

4. Please provide additional details regarding participant consent. In the ethics statement in the Methods and online submission information, please ensure that you have specified (1) whether consent was informed and (2) what type you obtained (for instance, written or verbal, and if verbal, how it was documented and witnessed). 

Response: As we now make clear in the manuscript, all participants signed a digital informed consent form before starting the online experiments (see p.7, lines 146-147, and p.14, lines 314-316, in the revised manuscript).

Response: We have made the data supporting the findings of this project openly available in the Open Science Framework (OSF) repository at https://doi.org/10.17605/OSF.IO/J8Q9A

Response: As requested, we moved the ethical statements from the introduction to the methods sections of Studies 1 and 2 (see p.6, lines 134-137; p.7, lines 146-147; p.13, lines 301-304; and p.14, lines 314-316, in the revised manuscript).

Response: We carefully reviewed our reference list and ensured it was complete and correct. A new citation was added to the manuscript (i.e., citation 28) to address comment no. 4 by Reviewer 1. 

 

Reviewer #1 Comments

1. Line 46, reference missing for the studies referring to adults behaving in a less prosocial manner when rejected by society.

Response: We thank the reviewer for pointing out the missing reference on line 46. A relevant reference has been added (see p.3, line 47, in the revised manuscript).

2. Page 4, line 74 to 78 is not clear to me. Please simplify this sentence or rephrase it. 

Response: Following the Reviewer’s comment, we rephrased the sentence to convey our argument better. Namely, instead of “ingroup bias,” we used the more accurate and descriptive term “ingroup favoritism.” We also added an elaborating sentence to explain it. The new section now reads: “However, an intergroup context does not automatically lead to a preference toward the ingroup (i.e., ingroup favoritism). For example, it is less likely to occur when the ingroup is of low status [21,22] or when the social norm strongly denounces discrimination [23]. In such circumstances, people are less likely to prefer members of their ingroup over outgroup members” (see p.4, lines 73-77, in the revised version of the manuscript). In addition, we moved the sentence that followed the above one to a new paragraph, as it begins to describe a new idea (see p.4, lines 78-81, in the revised version of the manuscript). We believe these changes simplify the text. See also our response to comment 4.

3. “According to these theories, a high degree of identification with one's ingroup increases positive one's ingroup evaluations and ingroup bias”, This sentence is unclear. Perhaps there is a slight mistake on the usage of the word “positive”.

Response: We agree with the Reviewer that this sentence is unclear and thank them for highlighting it. We amended the manuscript to clarify this sentence. Namely, we now write: “According to these theories, a high degree of identification with one's ingroup increases positive evaluations and preference of the ingroup (i.e., ingroup favoritism); however, a low degree of identification decreases positive evaluations and preference of the ingroup [(26,27])” (see p.4, lines 83-87, in the revised version of the manuscript).

4. “Our hypothesis that socially excluded people's decrease in prosocial behavior would be confined to fellow ingroup members who have rejected them gives rise to the idea that ingroup favoritism may be attenuated by people's sense of belonging to their group or community.” This argument has to be clearly presented and stregthen in the manuscript. As of now, the argument of having an intergroup context is weak.

Response: We thank the Reviewer for pointing out the need to clarify and strengthen our argument. To that end, we reviewed the literature again and found a highly relevant study on children that supports our argument that ostracism will reduce prosocial behavior toward the ingroup due to a decreased sense of belonging. By describing this new study’s results and how it supports our argument, we hope to articulate better and clarify our hypothesis (see pp.4-5, lines 87-95). Namely, we added the following paragraph to the manuscript:

Recent findings from a study conducted on fourth-grade school children support the hypothesis that ostracism will reduce prosocial behavior toward the ingroup but not the outgroup due to a reduced sense of belonging [28]. In this study, the authors found a strong positive correlation between popularity (i.e., the number of children who reported playing with each child during school breaks) and prosocial behavior toward the ingroup (i.e., sharing sweets). Namely, less popular children were less likely to share their sweets. The authors concluded that less popular children are those whose sense of belonging to their group is jeopardized. Hence, they are less likely to feel obligated to act prosaically towards their group.

5. “participants are assigned to groups based on an arbitrary criterion.” It would be nice to use existing ingroup and outgroup criterion, for example ethinicity, age as examples.

Response: This is indeed a very interesting idea that will also increase the ecological validity of the study and one that future studies should address. We now refer to this in the discussion section. Namely, we acknowledge that we operationalized group affiliation using the minimal group paradigm and that future studies should replicate our findings in real-world groups to increase the ecological validity of the findings (see p.20, lines 461-463). However, it is important to note that there is a consensus in the literature regarding the reliability and validity of the minimal group paradigm. That is, effects that were found using the minimal group paradigm should be found in real-world groups where group affiliation is stronger.

6. “(1) How much did you enjoy playing the ball tossing game? (2) How much did you feel a part of the group?, and (3) How much did you feel other players in the group appreciated you as a person?” Please explain how these questions are vaidating the manipulation of the games? If the were to answer 7 – very much, what would it mean and 1 not at all, what would that mean in terms of the manipulation of the game?

Response: Following the Reviewer’s comment, we now understand that the meaning of the manipulation check was unclear. Hence, we added a sentence to clarify this point. Namely, we now explain that the purpose of the three manipulation check questions was to confirm that participants assigned to the exclusion condition of the ball tossing game 1) enjoyed the game less; 2) felt less a part of their group; and 3) felt less appreciated by the other players in their group, compared to participants assigned to the inclusion condition. Namely, we expected participants in the exclusion condition to have a lower mean score on the three manipulation check questions than participants in the inclusion condition (see p.9, lines 187-192, in the revised version of the manuscript). 

 Indeed, in both studies, excluded participants reported enjoying playing the Cyberball game less, felt a weaker sense of belonging to their group, and felt less appreciated by the members of their group, than included participants (see p.10, lines 227-231, and pp.14-15, lines 332-337, in the revised version of the manuscript).

7. How can you be sure that the participants in Study 2 are not the same as in Study 1 from MTurk? Could it be a potential bias to the results if same participants signed up for both studies?

Response: We thank the reviewer for pointing out the need to clarify this point. Indeed, participants in Studies 1 and 2 were not the same. We made sure of this using built-in exclusion criteria in MTurk, allowing us to offer Study 2 only to participants that met the inclusion criteria and did not participate in Study 1. We now clarify this point in the text (see p.13, line 309, in the revised manuscript).

8. It would be neat to have two more studies replicating Study 1 and Study 2 in a different context or using a different game. This is to show robustness. In my opinion, two studies are not enough to show the robustness of the results.

Response: We completely agree with the Reviewer that our findings should be replicated in different contexts. More specifically, we intend to use a broader range of ostracism manipulation (e.g., fake sociometric feedback) and examine various prosocial behaviors (e.g., donating, volunteering, helping, and collaborating). We refer to this point as a limitation of the study on p.20, lines 463-465 in the revised version of the manuscript.

Despite the lack of such studies in the current manuscript, we hope the Reviewer sees the merit of our two novel studies, showing ostracism's effect on prosocial behavior in an inter-group context for the first time. We are currently working in several directions to expand the scope of our findings, and we hope that we will be able to publish our upcoming results in the next few years. 

9. What is the difference between the usage of words: ingroup vs outgroup and team member vs group members? Can’t they all be ingroup vs outgroup? This brings me to the next question, the comparison between ingroup and outgroup and being a team member in a bigger group is a bit confusing. This should be explain clearer.

Response: The terminology of “ingroup” and “outgroup” are used to describe the theoretical concepts, while “team member” and “group member” are both referring to the operationalization of ingroup. As we distinguish between ingroup members participants interacted with vs. ingroup members participants did not interact with, “team members” refer to the members of the ingroup that participants interacted with and that directly included or excluded them during the Cyberball game. Group members, on the other hand, refer to all the ingroup members, not just the team members with whom participants interacted during the Cyberball game. Following the Reviewer’s comment, we now see that these terms and their meanings were poorly defined. Therefore, we amended the manuscript to explain the usage of these terms better (see p.13, lines 292-296).

10. Please be consistent with the usage of the word Fig vs Figure in Figure caption.

Response: We thank the reviewer for noticing the inconsistency. Following PLOS ONE's style requirements, we changed “Figure” to “Fig” (see p.15, line 350, in the revised manuscript).

11. The manuscript is written well, the idea is also well thought of. However, the manuscript could improve with the help of a proofreader or streamlining the explanation in a clearer manner. This is important when the studies are complicated.

Response: Following the Reviewer’s request, the revised version of the manuscript was carefully proofread.

---

## [Editor Report · Decision Letter 1]

30 May 2023

Ostracism and sharing in an intergroup context

PONE-D-23-00987R1

Dear Dr. Authors,

We’re pleased to inform you that your manuscript has been judged scientifically suitable for publication and will be formally accepted for publication once it meets all outstanding technical requirements.

Kind regards,

Arslan Ayub

Academic Editor

PLOS ONE
---

## [Editor Report · Acceptance letter]

2 Jun 2023

PONE-D-23-00987R1 

Ostracism and sharing in an intergroup context 

Dear Dr. Zeev-Wolf:

I'm pleased to inform you that your manuscript has been deemed suitable for publication in PLOS ONE. Congratulations! Your manuscript is now with our production department. 

Kind regards, 

on behalf of

Dr. Arslan Ayub 

Academic Editor

PLOS ONE